# Yield, Quality, and Nitrogen Leaching of Open-Field Tomato in Response to Different Nitrogen Application Measures in Northwestern China

**DOI:** 10.3390/plants13070924

**Published:** 2024-03-22

**Authors:** Xinping Mao, Jialin Gu, Fang Wang, Kun Wang, Ruliang Liu, Yu Hong, Ying Wang, Fengpeng Han

**Affiliations:** 1State Key Laboratory of Soil Erosion and Dryland Farming on the Loess Plateau, College of Natural Resources and Environment, Northwest A & F University, Xianyang 712100, China; maoxinping_@nwafu.edu.cn (X.M.); wk20180910@163.com (K.W.); 2Institute of Agricultural Resources and Environment, Ningxia Academy of Agro-Forestry Science, Yinchuan 750002, China; berrywang@vip.sina.com (F.W.); ruliang_liu@126.com (R.L.); ouxuer333@163.com (Y.H.); wy6789668@163.com (Y.W.); 3National Agricultural Environment Yinchuan Observation and Experiment Station, Ningxia Academy of Agro-Forestry Science, Yinchuan 750002, China; 4Institute of Plant Nutrition, Resources and Environment, Beijing Academy of Agricultural and Forestry Sciences, Beijing 100097, China; 13681240815@163.com; 5Research Center on Soil & Water Conservation, Institute of Soil and Water Conservation, Chinese Academy of Sciences Ministry of Water Resources, Xianyang 712100, China

**Keywords:** agricultural non-point source pollution, nitrogen use efficiency, nitrogen leaching, open-field tomato, principal component analysis, quality

## Abstract

The overuse of fertilizers in open-field tomato leads to soil deterioration through nutrient leaching and increases the risk of agricultural non-point source contamination. Currently, the combined effects of different fertilization methods on soil nitrogen leaching and tomato production are still unclear. Therefore, the most effective fertilization method for open-field tomato should be discovered by examining how different fertilization methods affected tomato yield and quality, nitrogen use efficiency (NUE), and soil nitrogen leaching. Compared with CK (no fertilization), fertilization significantly increased the yield, total sugar (TS), total soluble solids (TSS), and vitamin C (vC) contents of fruits (*p* < 0.05), and OPT (optimal fertilization, controlled release nitrogen application, 240 kg ha^−1^) had the largest effect on increasing yield, quality, and net profit. However, when the fertilizer application rate reached 375 kg ha^−1^, these indices decreased. Nitrogen leaching concentrations, leaching amount, and titratable acids (TAs) increased with increased nitrogen application rates. Compared with other treatments, OPT reduced the total leaching amounts of total nitrogen (TN), nitrate nitrogen (NO_3_^−^-N), and ammonia nitrogen (NH_4_^+^-N) by 30.09–51.79%, 24.89–50.03%, and 30–65%, respectively. Principal component analysis (PCA) showed that OPT achieved the highest overall score in terms of yield, quality, and nitrogen leaching conditions. The partial least squares path modeling (PLS-PM) further reveals that applications of high amounts of nitorigen have a positive effect on soil nitrogen leaching. The amount of nitrogen leaching vegetatively affects tomato yield and quality, while plant uptake of nitrogen positively affects tomato production. These findings confirm the importance of using controlled-release fertilizers and reducing nitrogen inputs to control nitrogen leaching and enhance open-field tomato yields.

## 1. Introduction

Nitrogen is a crucial nutrient for plant growth and development. It also helps to boost crop yields and elevate the quality of agricultural output. Providing enough nitrogen to support high yields while enhancing soil fertility and reducing nitrogen loss to the environment is one of the key issues of agricultural nitrogen management [1,2]. Statistics show that the amount of nitrogen fertilizer used worldwide has been rising quickly. In 2006, it reached 89.45 × 10^6^ t, and in 2050, it is predicted to reach 85–260 × 10^6^ t [3]. According to Riddick et al. (2016) and Mueller et al. (2012), this extremely extensive nitrogen fertilizer application boosts agricultural output but also leaves behind significant amounts of nitrogen fertilizer residues that are not used by the crops and end up in the environment [4,5]. Because of rainfall and irrigation, the surplus of nutrients, which is prone to leaching into deep soil, results in fertilizer waste and nitrate pollution in surface and groundwater. Therefore, the study of nitrogen accumulation and leaching characteristics in soil has become a global focus [6,7,8].

China has become a major producer and consumer of vegetables, with an increase in vegetable-planting area from 15.24 million ha in 2000 to 21.49 million ha in 2020, which accounts 12.83% of the total crop planting area [9]. Open-field vegetables account for 35% of output value [10], playing an important role in vegetable production. However, the shortsighted pursuit of economic benefits by vegetable farmers frequently leads to daily water and fertilizer mismanagement. Nitrogen fertilizer input in vegetable fields can reach 1000 kg ha^−1^ or higher in one season, and the supply of water and fertilizer far exceeds the demand of the vegetable crops. China uses twice as much nitrogen fertilizer overall on open-field vegetables as it does on greenhouse vegetables [11]. Excessive fertilization not only wastes nitrogen fertilizer but also reduces crop yields [12]. Additionally, excess nitrogen leads to water eutrophication and groundwater nitrate pollution through surface runoff and soil percolation [13,14], which seriously damages the environment.

Tomato, which is native to South America, is now one of the main vegetables produced in China, although its production history is short. Open-field tomato is one of the main vegetables grown in the area irrigated by diversion of the Yellow River in Ningxia, a typical planting area in northwest China. Tomato is a typical vegetable with a shallow root system. The shallow root systems of vegetable crops, which are greatly influenced by natural factors such as rainfall, limit nitrogen absorption from the soil, often resulting in nitrogen leaching to the deep layer [15]. Meanwhile, in the pursuit of high yields, farmers apply large amounts of fertilizer, causing a high incidence of nitrogen leaching and shallow groundwater pollution in this area. In addition, tomatoes require a large quantity of water when they are growing. As irrigation occurs, soil-borne nitrogen molecules start to move downward with the water flow. Notably, nitrate ions (NO_3_^−^) are essential to this process due to their high water solubility and high sensitivity to being drawn out by water. These dissolved nitrogen molecules have the capacity to permeate the subsurface through soil pores and cracks, eventually seeping into groundwater reservoirs, leading to a buildup and enrichment of nitrogen within the groundwater system. Therefore, optimizing fertilization and reducing nitrogen leaching is important for improving tomato production and protecting the environment. At present, there is no detailed report on nitrogen leaching in open-field tomato culture in the area irrigated by the Yellow River in northwest China.

Controlled release fertilizers (CRFs) are specially designed to release active nutrients in a delayed and controlled manner according to the sequential nutrient requirements of plants. Therefore, they improve nutrient use efficiency and yield without causing nutrient loss [16]. In recent years, CRFs have been widely used in crop production as a new type of fertilizer that effectively improve the fertilizer utilization rate [17]. Studies have shown that CRFs can improve nutrient use efficiency and reduce environmental hazards without generally affecting production [18,19]. A leaching column study in California found that the leaching loss of soil nutrients, especially NO_3_^−^-N, within 16 weeks could be reduced by CRFs [20]. Furthermore, adding nitrogen synergists (urease inhibitors and nitrification inhibitors) with the goal of reducing the nitrogen dosage by 20% not only reduces nitrogen leaching loss but also has no effect on yield [21]. In the context of wheat–corn rotation, applying nitrogen efficiency enhancers can effectively improve nitrogen utilization efficiency and reduce the global warming potential (GHGI) [22]. These are effective leaching resistance control measures. However, within field tomato production, the temporal and spatial variations in soil nitrogen leaching due to different nitrogen application strategies and their corresponding impacts on tomato cultivation remain unclear. Therefore, investigating the beneficial impacts of various nitrogen application methods on reducing nitrogen loss, as well as the effects of these methods on tomato yield, quality, nitrogen use efficiency, and nitrogen leaching, will provide a scientific basis for reducing nitrogen loss and the development of environmentally friendly agricultural technology. In this study, it was hypothesized that optimal fertilization (controlled-release nitrogen) could reduce the risk of nitrogen leaching without reducing tomato yield and quality under different nitrogen application practices. The goals of this study were to (1) clarify the effects of different nitrogen application rates on the yield, quality, and nitrogen uptake of open-field tomato and (2) determine the characteristics and temporal and spatial rules of nitrogen leaching loss to provide a theoretical foundation for the sensible application of nitrogen fertilizer and prevention and control of non-point source pollution in the area irrigated by diversion of the Yellow River in northwestern China.

## 2. Results

### 2.1. Yield

Different nitrogen fertilization treatments resulted in varied fruit yield characteristics in open-field tomatoes (Figure 1a–c). Tomato yields were strongly impacted by nitrogen rate (*p* < 0.05) (Table 1). Each treatment showed a trend of gradually increasing yields with increasing nitrogen application rates until 300 kg ha^−1^, but decreased once the nitrogen supply reached 375 kg ha^−1^. OPT had the greatest yield (104.73 t ha^−1^), followed by N300 (92.30 t ha^−1^) and N375 (88.99 t ha^−1^). The lowest yield was 65.35 t ha^−1^ for CK. Compared with CK, the average yields increased by 60.25%, 41.24%, and 36.17 for OPT, N300, and N375, respectively. However, the N375 yield was 3.59% and 15.03% lower than those of the N300 and OPT treatments, respectively.

### 2.2. Nitrogen Use Efficiency (NUE), Agronomic Efficiency of Nitrogen (AEN), and Nitrogen Partial Factor Productivity (NPFP)

The total nitrogen uptake of tomato noticeably increased with increasing nitrogen application rates, and N150, N225, N300, N375, and OPT were significantly different from that of CK (*p* < 0.05). The highest total nitrogen uptake was 222.40 kg ha^−1^ in OPT (Table 2). The nitrogen uptake of stems was basically equivalent to that of fruits. The NUE of N150, N225, N300, N375, and OPT was 33.15%, 33.65%, 25.80%, 22.25%, and 34.95%, respectively. With an increase in nitrogen rate, NUE rose at first and subsequently fell. The nitrogen utilization rate was at its lowest (22.25%) when the nitrogen application rate reached 375 kg ha^−1^; the nitrogen utilization rate of OPT was the highest (34.95%), which was 57.08% higher than that of N375. AEN decreased in the following order: OPT > N225 > N150 > N300 > N375. NPFP decreased in the following order: N150 > OPT > N225 > N300 > N375. The results indicated that high nitrogen application rates did not improve the nitrogen absorption and use efficiency of tomato; however, OPT significantly increased the NUE, AEN, and NPFP of open-field tomato.

### 2.3. Concentrations of Leached Nitrogen

The concentrations of various forms of nitrogen (TN, NO_3_^−^-N and, NH_4_^+^-N) in leached water increased with increasing nitrogen application rates but decreased with increased irrigation times (Figure 2a–c). For each treatment, the highest leached TN, NO_3_^−^-N, and NH_4_^+^-N concentrations occurred after the first irrigation (17 May); the concentrations were 47.19–120.77 mg L^−1^, 38.51–84.63 mg L^−1^, and 0.57–4.48 mg L^−1^, respectively. The lowest leached nitrogen concentrations were observed after the final irrigation (15 July); the concentrations were 4.14–24.31 mg L^−1^, 2.22–14.54 mg L^−1^, and 0.13–0.27 mg L^−1^ for TN, NO_3_^−^-N and, NH_4_^+^-N, respectively. The leached concentration of NO_3_^−^-N was significantly higher than that of NH_4_^+^-N, and the TN and NO_3_^−^-N concentrations were significantly different after different irrigation periods (*p* < 0.05). Leached nitrogen was highest for N375 for different treatments, CK and OPT were the lowest in different periods, and OPT reduced the concentrations of different forms of nitrogen in the leached water.

### 2.4. Amount of Leached Nitrogen

TN, NO_3_^−^-N, and NH_4_^+^-N leaching mainly occurred after the first irrigation (17 May) (Figure 3a–c), and the amounts leached, respectively, were 4.27–10.63 kg ha^−1^, 3.47–8.10 kg ha^−1^, and 0.08–0.15 kg ha^−1^, which accounted for 44.94–48.8%, 45.18–47.34%, and 66.67–75.00% of the total leached losses of TN, NO_3_^−^-N, and NH_4_^+^-N, respectively, over the whole growth period of tomato. The leached amounts of different forms of nitrogen increased with increased nitrogen application rates but decreased with increased irrigation times. The leached amounts of TN, NO_3_^−^-N, and NH_4_^+^-N in each treatment were lowest after the fourth irrigation (15 July). Compared with other nitrogen treatments, the total leached losses of TN, NO_3_^−^-N, and NH_4_^+^-N were significantly reduced by OPT by 30.09–51.79%, 24.89–50.03%, and 30.00–65.00%, respectively.

### 2.5. Quality

Tomato quality indices responded differently to different fertilization treatments (Figure 4a–d). The TS and TSS contents were the highest in fruits in the OPT plot (3.54 g 100 g^−1^ and 6.98%, respectively), which were significantly higher than those obtained by other treatments (*p* < 0.05). Compared with CK, the TS and TSS contents in fruits treated with OPT increased by 32.96% and 22.91%, respectively. Increasing nitrogen application did not increase the TS and TSS contents in tomato fruit but had an inhibitory effect. When nitrogen treatment rates were raised, the TA content in tomato fruits first fell and subsequently rose. N375 had the highest TA content (4.31 g kg^−1^), while N225 and OPT had the lowest TA contents (3.22 g kg^−1^ and 3.44 g kg^−1^, respectively). OPT significantly reduced TA content and improved fruit quality. N375 generally had the highest vC content, and it decreased in the following order: N375 (32.95 mg 100 g^−1^) > N300 (32.35 mg 100 g^−1^) > OPT (31.10 mg 100 g^−1^) > N225 (30.23 mg 100 g^−1^) > CK (29.80 mg 100 g^−1^) > N150 (25.03 mg 100 g^−1^). High nitrogen application rates and controlled-release nitrogen fertilizer increased the vC content in fruits. In conclusion, the TS and TSS contents were highest in tomato fruits under OPT, while the TA content was lowest under OPT; therefore, controlled-release nitrogen fertilizer can significantly improve open-field tomato quality.

## 3. Discussion

### 3.1. Effects of Fertilization on Yield of Tomato

The pattern of nitrogen release from nitrogen fertilizer was the main factor affecting plant growth and output [23,24,25]. Because higher nitrogen levels promoted the crop’s physiological development and boosted its ability to absorb nutrients and water [26], tomato production in our research rose considerably (although not by more than 300 kg ha^−1^) with increasing nitrogen application. When nitrogen was provided at a rate of 375 kg ha^−1^, however, yields dramatically decreased (Table 1). Excessive nitrogen application significantly reduced the yield, which was consistent with studies reporting that when nitrogen application reaches a certain level, tomato yield decreases with continued increasing nitrogen application [27,28], which may be caused by increased osmotic pressure in the rhizosphere. The addition of controlled-release nitrogen fertilizer OPT maintained a continuous supply of inorganic nitrogen to the soil and significantly increased the yield of tomato. Compared with CK, the yield increase rate reached 60.25%, which was similar to the results of a study by Qu et al. (2020) [29]; they found that tomato yield increased by 25.2% compared with conventional fertilizer treatment when they used the controlled-release urea. This observation may be related to the nitrogen demand of tomato growth and the timing and intensity of nitrogen release and supply from the fertilizer [30]. Moreover, CRFs extend the retention time of nitrogen in the soil by regulating its release rate, thereby significantly mitigating the risk of nitrogen loss through various pathways [31]. This process enhances soil nutrient content and improves the nitrogen supply capacity of farmland. Adequate nitrogen supply contributes to prolonging the duration of plant dry matter accumulation and significantly enhances both nitrogen utilization efficiency and crop dry matter accumulation [32,33].

### 3.2. Effects of Nitrogen Rate on Nitrogen Use Efficiency (NUE), Agronomic Efficiency of Nitrogen (AEN), and Nitrogen Partial Factor Productivity (NPFP)

In this study, increasing the nitrogen application rate improved NUE, AEN, and NPFP. However, when the nitrogen dosage amounted to 375 kg ha^−1^, NUE, AEN, and NPFP decreased instead of increasing (Table 2). In several studies [34,35,36], it has been found that decreasing the usage amount of nitrogen fertilizer enhances NUE in vegetables grown in greenhouses. NUE was predicted to rise by 46% and 85%, respectively, if nitrogen input to greenhouse vegetables was reduced by 20% and 40%, according to Min et al. (2011) [36]. This could potentially be attributed to the fact that when the nitrogen content in the soil exceeds the plants’ growth requirements, the plants may struggle to efficiently absorb and utilize the excess nitrogen, leading to a decline in nitrogen use efficiency. On the other hand, excessive synthetic N input may destroy the function and structure of the soil microbial community [37] and inhibit nitrogen transformation and absorption. Paradoxically, this situation might elevate the risk of nitrogen leaching, consequently curtailing crop growth rather than enhancing it [38], while appropriate fertilizer application promotes nutrient absorption by crops. We found that the nitrogen uptake, NUE, and AEN were highest with OPT treatment (Table 2), and the NUE in OPT was 35.47% and 57.08% higher than in N300 and N375, respectively. The main reason for this may be the coating of the controlled-release fertilizer; it may have retarded the process of nitrogen leaching by blocking water migration through the coating, thus controlling the speed of nitrogen release and providing long-term absorption and utilization for tomato, which improved the NUE of tomato. This result was consistent with the research results of controlled-release fertilizer in potato, wheat, and rice [39,40,41,42]. The improvements in agronomic indices after the application of controlled-release nitrogen could be explained by higher fruit yields (Table 3 and Table 4).

### 3.3. Effects of Nitrogen Rate on Nitrogen Leaching

Compared with CK, both nitrogen leaching concentrations and leaching amounts increased with increasing nitrogen application rates (Figure 2 and Figure 3). Combined with the results in Table 3 and Table 4, this indicated that nitrogen application rates exceeding a certain threshold would no longer increase yield but reduce the NUE and increase nitrogen loss through leaching [43,44]. In contrast, soil environmental pollution caused by nitrogen leaching can be reduced by properly reducing nitrogen application rates. Zotarelli et al. (2009) proposed that when fertilizer input was reduced from 330 kg N ha^−1^ to 220 kg N ha^−1^, NO_3_^−^-N leaching was reduced by 50% [45]. By reducing the use of chemical fertilizers, soil environmental pollution caused by nitrogen leaching can be reduced, improving economic benefits and sustainable agricultural production [46].

Our study found that the leached amounts of different forms of nitrogen increased with increased nitrogen application rates but decreased with increased irrigation times (Figure 3). This could be explained by the fact that when nitrogen application rates rise, the enhanced N supply often leads to overgrowth of plant vegetative organs and an imbalance of nutrient elements in plant tissues [47]. However, with an increase in irrigation frequency, the residual nitrogen concentration in the soil gradually decreased, consequently leading to a reduction in the amount of nitrogen leaching. OPT significantly reduced the total leached amounts of TN, NO_3_^−^-N, and NH_4_^+^-N by 30.09–51.79%, 24.89–50.03%, and 30.00–65.00%, respectively. Controlled-release fertilizers can control the process and amount of nitrogen release, thus effectively limiting the loss of nitrogen to water. Similar to the results of research on greenhouse tomatoes, potatoes, corn, sweet peppers, and rice [18,48,49,50] (Figure 2 and Figure 3), the use of alternatively coated nitrogen fertilizers has been declared an effective strategy for reducing water pollution due to eutrophication [51]. Moreover, as of May 2023, the price of urea is USD 626 per ton [52]. The price of controlled-release urea is only approximately 20% higher than that of conventional urea. When reducing the application rate and frequency of fertilization, economic input is comparable to that of conventional fertilizers. Controlled-release urea has been widely adopted in China.

### 3.4. Effects of Nitrogen Application Measures on Fruit Quality Indicators

TS, TSS, TA, and vC are important indices of tomato fruit quality and flavor. Their contents are directly related to the nutritional value and flavor of tomato. Our study found that TS, TSS, and vC contents increased significantly with increasing nitrogen application rates of up to 300 kg ha^−1^. Proper nitrogen application significantly improves nitrogen absorption, thus contributing to improvements in photosynthetic activity and the creation of proteins [53]. The TS, TSS, and vC concentrations fell, but the TA content rose when nitrogen application rates were raised to 375 kg ha^−1^. Excessive nitrogen level prevents tomato fruits from absorbing nutrients but encourages the synthesis of amino acids from organic acids. However, since the creation of organic acids necessitates the conversion of sugar, this mechanism boosts sugar consumption while lowering sugar buildup [54,55,56]. Thus, an extremely high nitrogen application inhibited fruit TSS formation but increased TA content (Figure 4). The basis of vC synthesis is sugar. Accordingly, the decrease in TS could have prevented the synthesis of vC, which is similar to the discoveries of other investigations [57,58,59]. In our study, controlled-release nitrogen significantly increased fruit production (Table 1), TS, and TSS contents (Figure 4a,b) and decreased nitrogen leaching (Figure 2 and Figure 3), which may be because controlled release of nitrogen through a polymer coating reduced the rate of nitrogen leaching with water, reduced the rapid release of nitrogen in fertilizer, and met the nutrient requirements of tomato fruit stages as well as having a lower impact on the environment.

### 3.5. Quantitative Pathways from Soil Nitrogen Leaching to Tomato Growth

Three principal components with eigenvalues >1 were identified using PCA, and their total variance contribution ratio was 92.47%. The factor load and variance contribution rate are shown in Table 3. The variance contribution ratio of PC1 was 50.93%, which had a positive correlation with yield, TN leaching amount, NO_3_^−^-N leaching amount, NH_4_^+^-N leaching amount, TS, TSS, TA, and vC; these factors were positively correlated (*p* < 0.05) with a correlation coefficient of >0.7. The variance contribution ratio of PC2 was 27.41%, which was mainly affected by the positive influence of NH_4_^+^-N leaching and the negative influence of total sugar. Therefore, PC2 increased with increased NH_4_^+^-N leaching and decreased with increased TS. The variance contribution ratio of PC3 was 14.13%, which was positively impacted by TA; thus, PC3 increased with increased TA. Combined with the variance contribution rates of the three PCs, the comprehensive linear function of each treatment based on yield, nitrogen leaching loss, and quality could be written as follows:Z = 0.5093*PC1* + 0.2741*PC2* + 0.1413*PC3*,(1)
where Z is the comprehensive score of each treatment; *PC1* is the comprehensive score of *PC1*; *PC2* is the comprehensive score of *PC2*; and *PC3* is the composite score of *PC3*. The raw data were first standardized to eliminate dimensional effects. Then, the comprehensive score and ranking of each treatment were obtained by substituting them into Equation (1) (Table 3 and Table 4). The comprehensive ranking of OPT was the highest, with a comprehensive score of 0.94, and *PC1* and *PC3* were the highest; that is, the yield and quality (TSS, vC, and TS) had the highest values. With increased nitrogen leaching loss, the overall treatment ranking is reduced. Greater fruit production may be used to explain the significant improvement in agronomical indices seen when applying coated fertilizer (Table 1).

PCA (Figure 5a) showed the overall impacts of diverse nitrogen application rates on nitrogen leaching and plant characteristics. The PCA (Figure 5a,b) showed that 78.3% of variability was explained by component 1 (50.9%) and component 2 (27.4%). TN leaching and NO_3_^−^-N leaching were highly correlated with TA, which were the main factors promoting TA formation. NH_4_^+^-N leaching was highly negatively correlated with TS. PCA also revealed significant differences between OPT and other treatment methods (Figure 5b).

The path analysis was conducted using partial least squares path modeling (PLS-PM) to examine the causal relationships and strengths among soil nitrogen leaching, nitrogen utilization efficiency, tomato yield, and quality. Figure 6 reveals that both NO_3_^−^-N, and NH_4_^+^-N leaching have negative impacts on NUE and tomato yield, with coefficients of −4.029, −0.571, −0.895, and −0.012, respectively. Conversely, there is a positive influence observed between nitrogen utilization efficiency, yield, and quality. Notably, nitrogen utilization efficiency significantly contributes, with a coefficient of 0.645, towards the overall yield.

Based on the discussion above, fertilization measures are essential factors influencing soil nitrogen leaching. Increasing the application of nitrogen fertilizer can lead to higher leaching of total nitrogen, nitrate nitrogen, and ammonium nitrogen in the soil, thus increasing the risk of nitrogen leaching. However, using controlled-release nitrogen fertilizers to optimize fertilization can effectively reduce soil nitrogen leaching without compromising crop yield. This may be attributed to the ability of controlled-release fertilizers to regulate the rate of nitrogen release, thereby lowering nitrogen concentrations in the soil. Additionally, their granular or slow-release film forms can improve soil water retention and maintain soil structure, reducing water runoff and soil erosion, which helps minimize nitrogen leaching.

### 3.6. Effect of Fertilization Level on Tomato Profit

Yield and net profit are the most crucial concerns for open-field tomato growers. In our study, compared to the CK, all treatments resulted in increased yields (Table 1) and enhanced economic benefits of tomato cultivation (Table 5). Under equivalent field management practices, the OPT treatment exhibited the highest economic benefits, reaching 55.96 K CNY ha^−1^, which represents a 44.51% increase compared to conventional fertilization (N375). This indicates that nitrogen application measures not only influence tomato yield and nitrogen utilization efficiency (Table 2) but also have the potential to impact the local environment and tomato economic returns.

## 4. Materials and Methods

### 4.1. Experimental Site

The field plot experiment (106°22′14″ E, 38°47′62″ N) was conducted from 2020 to 2021 at the Experimental Base of the Ningxia Academy of Agriculture and Forestry Sciences, Wanghong Town, Yongning County, Ningxia Province, China. The climate is middle-temperate and arid. The average yearly temperature is 8.7 °C, and the annual precipitation, which ranges from 180–220 mm, mostly falls in June, July, and September. The mean annual evaporation is 1800 mm, and the frost-free period is 143–160 days long. The main crop in this area contains paddy, maize, wheat, tomatoes, and peppers frequently irrigated by Yellow River water due to the limited precipitation. The primary soil type in this area is anthropogenic–alluvial soil with coarse soil texture, characterized by a high sand content (>50%) and low clay content (<5%); the fundamental physicochemical characteristics (0–100 cm) of the tested soil are shown in Table 6.

### 4.2. Experimental Design and Management

In this experiment, six treatments were designed: (1) CK (no fertilization); N150 (nitrogen fertilizer dosage of 150 kg ha^−1^); N225 (nitrogen fertilizer dosage of 225 kg ha^−1^); N300 (nitrogen fertilizer dosage of 300 kg ha^−1^); N375 (farmer’s experience-based fertilizer application strategy, nitrogen fertilizer dosage of 375 kg ha^−1^); and OPT (optimal fertilization involving controlled-release nitrogen application, 240 kg ha^−1^). For tomato, the N, P, and K fertilizers were applied as urea (N, 46%), heavy superphosphate (P_2_O_5_, 46%), and potassium chloride (KCl, 60%), respectively. The phosphate and potassium were applied before transplantation as base fertilizers. For N150, N225, N300, and N375, 75% of the fertilizer was applied as base fertilizer before transplantation, and the remaining 25% was applied in the flowering stage (15 June). For OPT treatment with controlled-release nitrogen fertilizer (N content of 44%), 100% of the fertilizer was applied as base fertilizer before transplantation. The base fertilizers were applied by broadcasting and subsequently covered with soil. The topdressing fertilizers were applied in the planting hole. Urea, heavy superphosphate, and potassium chloride were purchased from fertilizer companies, namely Sinopec (China Petrochemical Corporation, Beijing, China), the Yunnan Yuntianhua Group, and the China National Agricultural Means of Production Group, respectively. The controlled-release nitrogen fertilizer was obtained from the Shandong Yantai Agricultural Capital Sun Fertilizer Company (Yantai, Shandong Province, China).

Prior to the experiment, a survey was conducted to obtain information on the conventional fertilization practices of farmers. Referring to the local farmers’ application rates of phosphorus and potassium fertilizers, phosphorus and potassium application rates were set for the N150, N225, N300, and N375 treatments. Soil testing was conducted before tomato transplanting, revealing high levels of available phosphorus and potassium in the soil surface layer, meeting the growth requirements of tomatoes. Based on the target yield, NPK fertilizer application rates for the OPT treatment were designed. OPT, which utilized controlled-release compound fertilizer with a coating material that enabled the slow release of nutrients, met the seasonal fertilizer demands of the crops. Specific fertilizer application rates are shown in Table 7.

The tomato variety tested was “Fenguo HS130”, and the planting method consisted of seedling cultivation and transplantation. The field was prepared on 20 April by applying base fertilizer. Transplantation was conducted on 9 May, top dressing occurred on 15 June, harvesting was conducted on 26 September, and fruits were collected a total of six times during the ripening period. The plot area, arranged randomly, was 60 m^2^ (6 m × 10 m), and each treatment was repeated three times. The length was measured from east to west, and the width was measured from north to south. The width of the field ridge was 50 cm, and the height was 30 cm. Each two rows of districts shared a 1.5 m wide irrigation canal and a 1 m wide guard row. The field ridge was buried with black, long-lasting plastic film from 30 cm above the soil surface to a soil depth of 100 cm. Each plot was separated to prevent lateral infiltration between them. Two 100 cm wide beds located 50 cm apart were created in each plot before planting, and two rows were planted in each bed. The density was 29,340 plants ha^−1^, and the plants were 45 cm apart.

### 4.3. Sampling and Measurements

Before the test period began, a water suction cup was planted in the center area of each plot at a depth of 0.9 m, with an area of 1.2 m^2^, and was used to collect soil water samplers. In order to ensure the precision of the field sampling apparatus, the first year of the experiment was dedicated to allowing the leaching pans to naturally compact, with only field experiments being arranged. The results presented in this article are based on research data collected in the second year following the experimental setup. The leached water was collected about 7 days after each irrigation, and its volume was measured. A 500 mL sample was placed in a sample bottle, taken back to the laboratory, and stored at 4 °C; determinations were completed within 24 h. From the beginning of the tomato harvest period to the end, the cumulative yield was calculated by counting plots, and the fruits were collected separately. The fruits were graded in the field as extra-large (greater than 7.00 cm), large (6.35–7.06 cm), medium (5.72–6.43 cm), and unmarketable (cull) fruit according to USDA standards [60]. The average fruit size was medium (5.72–6.43 cm); thus, four medium-sized and colored full-grown tomato fruits were randomly chosen from each plot at the full fruit stage and homogenized in a blender to measure the total soluble solids (TSS), total sugar (TS), vitamin C (vC), and titratable acid (TA). A portable refractometer (PR-32; ATAGO, Tokyo, Japan) was used to measure TSS. The 2,6-dichlorophenolate indoxol sodium salt solution was used for the detection of vC, and phenolphthalein–NaOH titration was employed for the determination of TS and TA [61]. In the final harvest period, five plants and fruits were randomly selected from each plot. After 30 min in an oven at 45 °C, they were dried at 70 °C to a constant weight; the dried materials were used to determine their total nitrogen contents and to calculate the aboveground biomass and nitrogen use efficiency of tomato. Total nitrogen and ammonia nitrogen and nitrate nitrogen were determined in leached water samples using potassium persulfate oxidation–ultraviolet spectrophotometry and flow injection analysis, respectively [62].

### 4.4. Data Processing and Statistical Analysis

Nitrogen use efficiency (NUE) was computed with the following formula [63]:*NUE* (*%*) = (*N_t_* − *N_c_*)/*NF* × 100, (2)
where *N_t_* is the total N uptake in the fertilized plot (kg ha^−1^), *N_c_* is the total N uptake in the CK plot (kg ha^−1^), and *NF* is the amount of nitrogen fertilizer applied (kg ha^−1^).

The agronomic efficiency of nitrogen (AEN) was computed with the following formula [64]:*AEN* (kg kg^−1^) = (*Y_x_* − *Y*_0_)/*NF*, (3)
where *Y_x_* is the yield at a nitrogen fertilizer level of X (kg ha^−1^), *Y*_0_ is the yield with no nitrogen fertilizer (kg ha^−1^), and *NF* is the amount of nitrogen fertilizer applied (kg ha^−1^).

Nitrogen partial factor productivity (NPFP) was computed with the following formula [65]:*NPFP* = *Y*/*NF*, (4)
where *Y* is the total fruit yield (kg ha^−1^) and *NF* represents the quantity of nitrogen fertilizer used (kg ha^−1^), i.e., the total nitrogen fertilizer used throughout a tomato-growing season.

Nitrogen leaching loss (*P*) was calculated using the following equation:(5)$$P=\sum_{i=1}^{n}Ci\times Vi$$
where *Ci* is the concentration of nitrogen in the *i*th leached water (mg L^−1^), and *Vi* is the volume of leached water at the *i*th time (L).

Data processing and plotting were performed using Excel 2010 (Microsoft Corp., Redmond, WA, USA) and Origin 2022b (Origin Lab Ltd., Guangzhou, China), and the analyses of variance and principal component analyses (PCA) were conducted using SPSS 20.0 (SPSS Inc., Chicago, IL, USA). The correlation between soil nitrogen leaching and tomato N uptake, yield, and quality was verified using the partial least squares path model (PLS-PM) with R 4.3.1. The least significant difference (LSD) test was used to compare the means with a 5% threshold of significance.

## 5. Conclusions

Fertilization could significantly increase tomato yield, NUE, TS, TSS, and vC contents (*p* < 0.05); however, as the nitrogen fertilizer dosage increased to 375 kg ha^−1^, these indices started to decline. Compared with other fertilization measures, application of OPT (N 240–P_2_O_5_ 90–K_2_O 45 kg ha^−1^) achieved a good compromise between the yield, NUE, and fruit quality of open-field tomato; thus, OPT proved able to improve fruit quality and save fertilizer. OPT also effectively reduced the leached concentrations and amounts of TN, NO_3_^−^-N, and NH_4_^+^-N at different growth stages of tomato, especially reducing the pollution risk of NO_3_^−^-N leaching into groundwater, thereby alleviating the risk of non-point-source pollution caused by fertilizer leaching. In addition, tomatoes produced with OPT have improved taste and effects on human health due to their high TS and vC contents. The PCA and PLS-PM comprehensive evaluation, including tomato fruit yield, quality, and nitrogen leaching loss, indicated that OPT was the best nitrogen treatment. These results are of great significance for the improvement of nitrogen fertilizer management in open-field vegetables and for the implementation of rational fertilization systems in northwest China and other parts of the world. Going forward, we will focus on studying the relationship between irrigation and fertilizer levels, including more grade levels, to better estimate the input rate of OPT.

## Figures and Tables

**Figure 1 plants-13-00924-f001:**
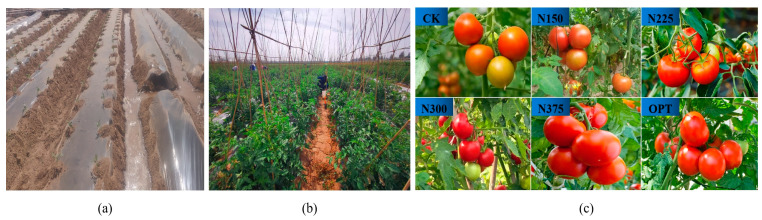
Planting and growth of open-field tomato. (**a**), tomato transplanting, (**b**), tomato flowering stage, (**c**), the full fruit period of different treatments. CK refers to 0 fertilization; N150, N225, N300, and N375 refer to 150, 225, 300, and 375 kg N ha^−1^, respectively; and OPT refers to optimal fertilization (controlled-release nitrogen fertilization, 240 kg N ha^−1^). The same applies below.

**Figure 2 plants-13-00924-f002:**
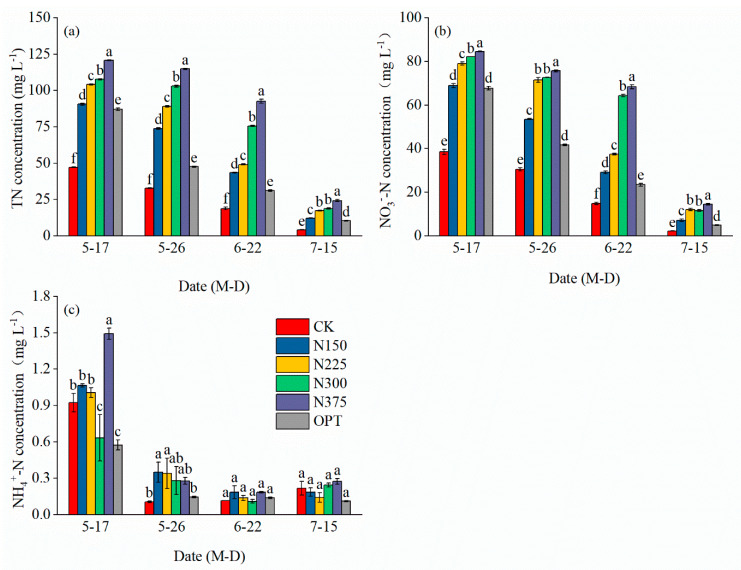
Comparison of leached concentrations of TN, NO_3_^−^-N, and NH_4_^+^-N under different treatments. (**a**), total nitrogen (TN) concentration (mg L^−1^), (**b**), nitrate nitrogen (NO_3_^−^-N) concentration (mg L^−1^), (**c**), ammonia nitrogen (NH_4_^+^-N) concentration (mg L^−1^). The lowercase letter(s) over the bars indicate significant differences according to LSD multiple range tests at *p* < 0.05, and the error bars represent standard deviation or standard error (n = 3).

**Figure 3 plants-13-00924-f003:**
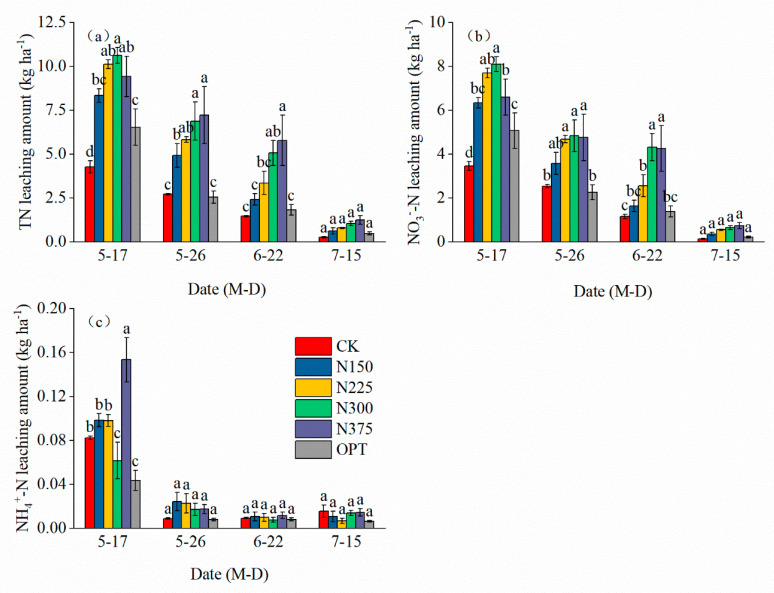
Comparison of leached amounts of TN, NO_3_^−^-N, and NH_4_^+^-N under different treatments. (**a**), total nitrogen (TN) leaching amount (kg ha^−1^), (**b**), nitrate nitrogen (NO_3_^−^-N) leaching amount (kg ha^−1^), (**c**), ammonia nitrogen (NH_4_^+^-N) leaching amount (kg ha^−1^). The lowercase letter(s) over the bars indicate significant differences according to LSD multiple range tests at *p* < 0.05, and the error bars represent standard deviation or standard error (n = 3).

**Figure 4 plants-13-00924-f004:**
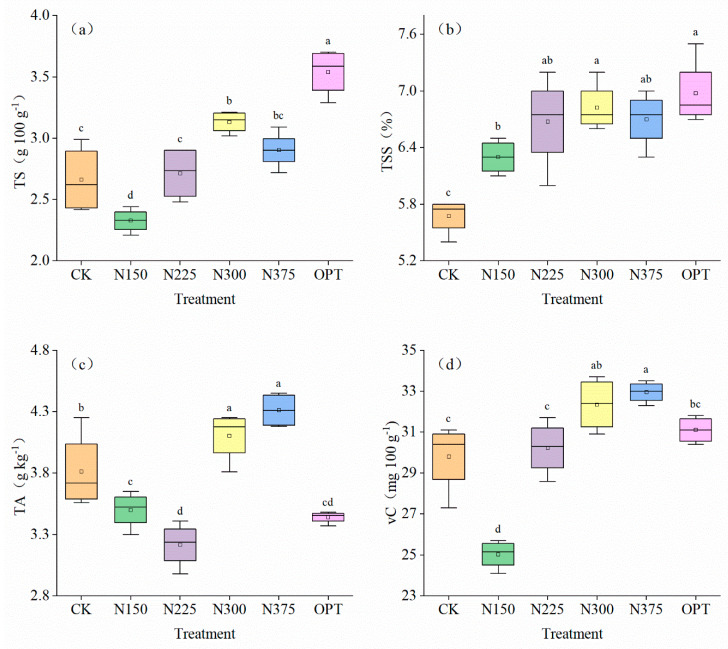
Comparison of tomato quality under different treatments. (**a**), total sugar (TS) content (g 100 g^−1^), (**b**), total soluble solids (TSS) content (%), (**c**), titratable acid (TA) content (g kg^−1^), (**d**), vitamin C (vC) content (mg 100 g^−1^). The lowercase letter(s) over the bars indicate significant differences according to LSD multiple range tests at *p* < 0.05, and the error bars represent standard deviation or standard error (n = 3). TS, TSS, TA, and vC represent total sugar (g 100 g^−1^), total soluble solids (%), titratable acid (g kg^−1^), and vitamin C (mg 100 g^−1^) in the tomato fruits.

**Figure 5 plants-13-00924-f005:**
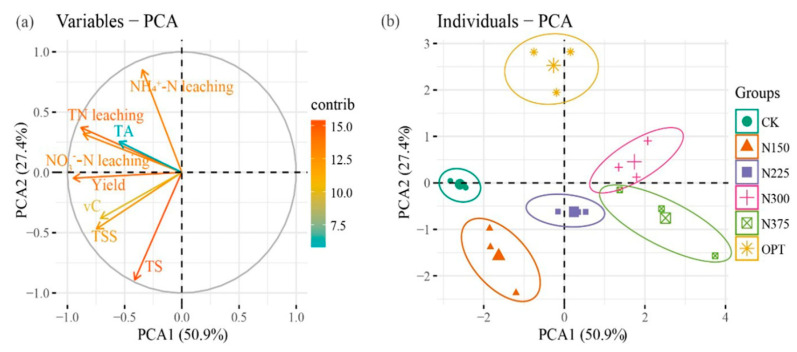
Principal component analysis (PCA1, PCA2) showing trait vectors (yield, total nitrogen leaching, nitrate leaching, ammonium leaching, total sugar, total soluble solids, titratable acids, and vitamin C, as indicated by arrows). Arrows indicate eigenvectors representing the strength (given by the length of the vector) and direction of the trait relative to the first two principal components (PCA1, PCA2) (**a**). The circles enclose those variables that fall into the same cluster (95% confidence level) (**b**). TN leaching, total nitrogen leaching; NO_3_^−^-N leaching, nitrate leaching; NH_4_^+^-N leaching, ammonium leaching; TS, total sugar; TSS, total soluble solids; TA, titratable acid; vC, vitamin C.

**Figure 6 plants-13-00924-f006:**
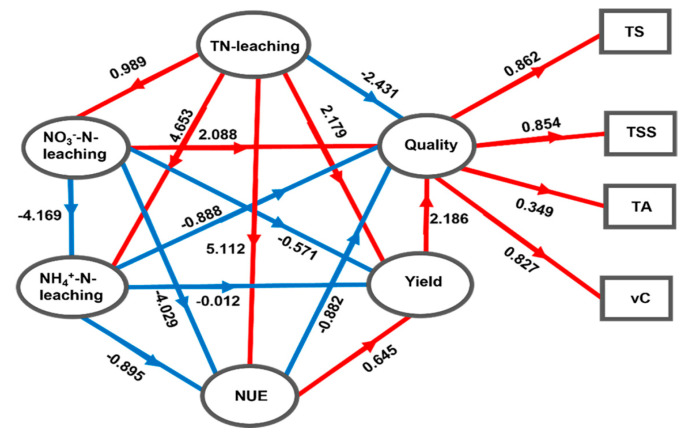
Results of partial least squares path modeling (PLS-PM). The causal relationships between six variables are depicted by arrows with path coefficients (red and blue arrows indicate positive and negative effects, respectively). The values above the arrows in the model represent the weights of the measured variables.

**Table 1 plants-13-00924-t001:** Differences in average yields under different treatments.

Treatment	Fruit Yield (kg ha^−1^)			Average Yield	Rate of Growth
	Trial I	Trial II	Trial III	(kg ha^−1^)	(%)
CK	60,239.68	67,296.70	68,520.09	65,352.16 ± 44.69 e	
N150	80,570.70	78,150.57	77,927.23	78,882.83 ± 14.66 d	20.70
N225	87,307.70	88,297.75	87,981.07	87,862.17 ± 5.06 c	34.44
N300	90,862.88	92,109.61	93,931.36	92,301.28 ± 15.43 b	41.24
N375	87,517.71	90,354.52	89,094.45	88,988.89 ± 14.21 bc	36.17
OPT	103,357.83	107,482.04	103,341.83	104,727.23 ± 23.86 a	60.25

Note: The letters (a, b, c, d, and e) following the average yield values represent significant differences at *p* < 0.05.

**Table 2 plants-13-00924-t002:** Differences in nitrogen use efficiency (NUE), agronomic efficiency of nitrogen (AEN), and nitrogen partial factor productivity (NPFP) under different treatments.

Treatment	Nitrogen Uptake by Stems	Nitrogen Uptake by Fruit	Total Nitrogen Uptake	NUE	AEN	NPFP
	kg ha^−1^	kg ha^−1^	kg ha^−1^	%	kg kg^−1^	kg kg^−1^
CK	66.19	72.34	138.53 ± 18.44 c	--	--	--
N150	86.44	101.82	188.26 ± 7.97 b	33.15	90.2	525.89
N225	109.72	104.51	214.23 ± 3.61 a	33.65	100.04	390.5
N300	121.89	94.03	215.92 ± 16.29 a	25.80	89.83	307.67
N375	109.14	112.84	221.98 ± 15.27 a	22.25	63.03	237.30
OPT	128.30	94.10	222.40 ± 21.91 a	34.95	164.06	436.36

Note: The letters (a, b and c) following the total nitrogen uptake values represent significant differences at *p* < 0.05.

**Table 3 plants-13-00924-t003:** Principal component factor loads and variance contribution rates.

Factor	Factor Loading		
	PC1	PC2	PC3
Yield	0.949	−0.048	−0.129
TN leaching amount	0.883	0.372	−0.211
NO_3_^−^-N leaching amount	0.861	0.319	−0.245
NH_4_^+^-N leaching amount	0.345	0.851	0.103
TS	0.412	−0.893	0.165
TSS	0.747	−0.469	−0.443
TA	0.548	0.254	0.736
vC	0.711	−0.382	0.483
Eigenvalue	1.074	2.193	1.131
Variability (%)	50.93	27.41	14.13
Cumulative %	50.93	78.34	92.47

Extraction method: principal component analysis. Three components were extracted.

**Table 4 plants-13-00924-t004:** Comprehensive scores for each treatment.

Treatment	PC1	PC2	PC3	Synthesis Score	Comprehensive Ranking
CK	−1.77588	−0.43719	0.65672	−1.82848	6
N150	−0.05573	−1.17492	−1.087	−0.63292	5
N225	0.68644	−0.19766	−0.97118	0.468866	3
N300	0.66473	0.52942	0.54525	0.933287	2
N375	−0.398	1.71009	−0.51532	0.121568	4
OPT	0.87844	−0.42975	1.37153	0.93768	1

Note: CK refers to 0 fertilization; N150, N225, N300, and N375 refer to 150, 225, 300, and 375 kg N ha^−1^, respectively; and OPT refers to optimal fertilization (controlled-release nitrogen fertilization, 240 kg N ha^−1^).

**Table 5 plants-13-00924-t005:** Effect of fertilization level on tomato profit.

Treatment	Increased Yield (kg ha^−1^)	Profit Increment (K CNY ha^−1^)	Fertilization Investment (K CNY ha^−1^)	Net Profit (K CNY ha^−1^)
CK	-	-	-	-
N150	13,530.67	19.62	2.71	16.91
N225	22,510.01	32.64	2.88	29.76
N300	26,949.12	39.08	3.05	36.03
N375	23,636.73	34.27	3.22	31.05
OPT	39,375.07	57.09	1.13	55.96

Note: Fertilizer price is urea 2.25 CNY kg^−1^, P_2_O_5_ 4.4 CNY kg^−1^, and KCl 4.4 CNY kg^−1^. Tomato price is 1.45 CNY kg^−1^.

**Table 6 plants-13-00924-t006:** Physical and chemical properties of soil.

Soil Depth (cm)	Bulk Density (g cm^−3^)	Total Porosity (%)	Total Salt Content (g kg^−1^)	Organic Matter (g kg^−1^)	Total Nitrogen (g kg^−1^)	Available Nitrogen (mg kg^−1^)
0–20	1.36	48.7	0.49	13.74	1.01	38.66
20–40	1.36	48.8	0.40	8.71	0.85	26.98
40–60	1.53	42.3	0.39	5.26	0.40	25.12
60–80	1.64	39.0	0.35	4.41	0.31	24.31
80–100	1.44	45.4	0.31	3.15	0.29	23.58

**Table 7 plants-13-00924-t007:** Amount of fertilizer applied for each treatment (kg ha^−1^).

Treatment	Common N	Control-Released N	P_2_O_5_	K_2_O
CK	0	0	0	0
N150	150	0	360	180
N225	225	0	360	180
N300	300	0	360	180
N375	375	0	360	180
OPT	0	240	90	45

Note: CK refers to fertilization; N150, N225, N300, and N375 refer to 150, 225, 300, and 375 kg N ha^−1^, respectively; and OPT refers to optimal fertilization (controlled-release nitrogen fertilization, 240 kg N ha^−1^). The same applies below.

## Data Availability

Data are contained within the article.

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
