# Peer review of "Yield, Quality, and Nitrogen Leaching of Open-Field Tomato in Response to Different Nitrogen Application Measures in Northwestern China"

_plants, 2024, doi:10.3390/plants13070924_

Round 1

Reviewer 1 Report

Comments and Suggestions for Authors

I carefully reviewed the manuscript entitled "Yield, quality, and nitrogen leaching of open-field tomato in response to different nitrogen application measures in northwestern China", by Xinping Mao and colleagues. The manuscript is well written and addresses an important issue in agriculture today, specifically soil pollution through the application of nitrogen-based fertilizers, and the effect of fertilizers, applied in different ways, on the quality of the obtained production.

The illustration is adequate and of appropriate quality, the explanations of the figures are correctly written. However, if the authors have at least one image (or more) of the experimental fields and the general appearance of the plants (if it were possible from several analyzed variants), their insertion at the beginning of the Results chapter would be more than welcome.

Below the authors can find some suggestions for improving the manuscript.

Line 24: For the abbreviation OPT, the explanation must be given the first time it appears in the text.

Line 45: "1 Tg = 1012 g" - the abbreviation must be correctly explained, Tg means tetragram or 1012 grams.

Lines 157-158: do you consider that the 50 cm deep plastic film is sufficient to eliminate infiltrations between the experimental fields? I am asking the question considering that nitrogen fertilizer often contaminates groundwater tables, which are at much greater depths.

Lines 391-392: "In our study, controlled release nitrogen significantly increased fruit TS, TSS, and vC contents and decreased TA content" - however, Figure 2 shows the fact that vitamin C (mg 100 g-1) is not higher in variant OPT compared to the last two variants that use nitrogen fertilizer (N300, N375); anyway, this is probably a small disadvantage, compared to the advantages related to fruit quality (increased production and total sugar quantity), as well as with the lower impact on the environment.

Line 494: References can be improved by adding newer papers that investigate similar topics. For example: Li, T.; Cui, J.; Guo, W.; She, Y.; Li, P. The Influence of Organic and Inorganic Fertilizer Applications on Nitrogen Transformation and Yield in Greenhouse Tomato Cultivation with Surface and Drip Irrigation Techniques. Water 2023, 15, 3546. https://doi.org/10.3390/w15203546.

Also this paper may be of interest for the current study: Li, T.; Cui, J.; Guo, W.; She, Y.; Li, P. The Influence of Organic and Inorganic Fertilizer Applications on Nitrogen Transformation and Yield in Greenhouse Tomato Cultivation with Surface and Drip Irrigation Techniques. Water 2023, 15, 3546. https://doi.org/10.3390/w15203546

Reviewer 2 Report

Comments and Suggestions for Authors

Comments on the Quality of English Language

Reviewer 3 Report

Comments and Suggestions for Authors

Please find enclosed my detailed review.

Round 2

Reviewer 1 Report

Comments and Suggestions for Authors

From my point of view, the authors made the changes suggested in the first round of revision. Consequently, the manuscript can be considered for publication in its current form.

Reviewer 2 Report

Comments and Suggestions for Authors

The manuscript is much improved after revision. I do not have further comments.

Reviewer 3 Report

Comments and Suggestions for Authors

The Materials and methods is currently between Discussion and Conclusion chapter. I suggest that it be placed back before Results.